# A Signal Propagation Perspective for Pruning Neural Networks at Initialization

**Namhoon Lee**[1], **Thalaiyasingam Ajanthan**[2], **Stephen Gould**[2], **Philip H. S. Torr**[1]

[1]University of Oxford
[2]Australian National University
[1]{namhoon,phst}@robots.ox.ac.uk
[2]{thalaiyasingam.ajanthan, stephen.gould}@anu.edu.au

## Abstract

Network pruning is a promising avenue for compressing deep neural networks. A typical approach to pruning starts by training a model and then removing redundant parameters while minimizing the impact on what is learned. Alternatively, a recent approach shows that pruning can be done at initialization prior to training, based on a saliency criterion called connection sensitivity. However, it remains unclear exactly why pruning an untrained, randomly initialized neural network is effective. In this work, by noting connection sensitivity as a form of gradient, we formally characterize initialization conditions to ensure reliable connection sensitivity measurements, which in turn yields effective pruning results. Moreover, we analyze the signal propagation properties of the resulting pruned networks and introduce a simple, data-free method to improve their trainability. Our modifications to the existing pruning at initialization method lead to improved results on all tested network models for image classification tasks. Furthermore, we empirically study the effect of supervision for pruning and demonstrate that our signal propagation perspective, combined with unsupervised pruning, can be useful in various scenarios where pruning is applied to non-standard arbitrarily-designed architectures.

## 1 Introduction

Deep learning has made great strides in machine learning and been applied to various fields from computer vision and natural language processing, to health care and playing games (LeCun et al., 2015). Despite the immense success, however, it remains challenging to deal with the excessive computational and memory requirements of large neural network models. To this end, lightweight models are often preferred, and *network pruning*, a technique to reduce parameters in a network, has been widely employed to compress deep neural networks (Han et al., 2016). Nonetheless, designing pruning algorithms has been often purely based on ad-hoc intuition lacking rigorous underpinning, partly because pruning was typically carried out after training the model as a post-processing step or interwoven with the training procedure, without adequate tools to analyze.

Recently, Lee et al. (2019) have shown that pruning can be done on randomly initialized neural networks in a single-shot prior to training (*i.e.*, *pruning at initialization*). They empirically showed that as long as the initial random weights are drawn from appropriately scaled Gaussians (*e.g.*, Glorot & Bengio (2010)), their pruning criterion called connection sensitivity can be used to prune deep neural networks, often to an extreme level of sparsity while maintaining good accuracy once trained. However, it remains unclear as to why pruning at initialization is effective, how it should be understood theoretically and whether it can be extended further.

In this work, we first look into the effect of initialization on pruning, and find that initial weights have critical impact on connection sensitivity, and therefore, pruning results. Deeper investigation shows that connection sensitivity is determined by an interplay between gradients and weights. Therefore when the initial weights are not chosen appropriately, the propagation of input signals into layers of

these random weights can result in saturating error signals (*i.e.*, gradients) under backpropagation, and hence unreliable connection sensitivity, potentially leading to a catastrophic pruning failure.

This result leads us to develop a signal propagation perspective for pruning at initialization, and to provide a formal characterization of how a network needs to be initialized for reliable connection sensitivity measurements and in turn effective pruning. Precisely, we show that a sufficient condition to ensure faithful[1] connection sensitivity is *layerwise dynamical isometry*, which is defined as all singular values of the layerwise Jacobians being concentrated around 1. Our signal propagation perspective is inspired by the recent literature on dynamical isometry and mean field theory (Saxe et al., 2014; Poole et al., 2016; Schoenholz et al., 2017; Pennington et al., 2017), in which the general signal propagation in neural networks is studied. We extend this result to understanding and improving pruning at initialization.

Moreover, we study signal propagation in the pruned sparse networks and its effect on trainability. We find that pruning neural networks can indeed break dynamical isometry, and hence, hinders signal propagation and degrades the training performance of the resulting sparse network. In order to address this issue, we propose a simple, yet effective data-free method to recover the layerwise orthogonality given the sparse topology, which in turn improves the training performance of the compressed network significantly. Our analysis further reveals that in addition to signal propagation, the choice of pruning method and sparsity level can influence trainability in sparse neural networks.

Perfect layerwise dynamical isometry cannot always be ensured in the modern networks that have components such as ReLU nonlinearities (Pennington et al., 2017) and/or batch normalization (Yang et al., 2019). Even in such cases, however, our experiments on various modern architectures (including convolutional and residual neural networks) indicate that connection sensitivity computed based on layerwise dynamical isometry is robust and consistently outperforms pruning based on other initialization schemes. This indicates that the signal propagation perspective is not only important to theoretically understand pruning at initialization, but also it improves the results of pruning for a range of networks of practical interest.

Furthermore, this signal propagation perspective for pruning poses another important question: how informative is the error signal computed on randomly initialized networks, or can we prune neural networks even without supervision? To understand this, we compute connection sensitivity scores with different unsupervised surrogate losses and evaluate the pruning results. Interestingly, our results indicate that we can in fact prune networks in an unsupervised manner to extreme sparsity levels without compromising accuracy, and it often compares competitively to pruning with supervision. Moreover, we test if pruning at initialization can be extended to obtain architectures that yield better performance than standard pre-designed architectures with the same number of parameters. In fact, this process, which we call *neural architecture sculpting*, compares favorably against hand-designed architectures, taking network pruning one step further towards neural architecture search.

## 2 PRELIMINARIES

**Pruning at initialization**.    The principle behind conventional approaches for network pruning is to find unnecessary parameters, such that by eliminating them the complexity of the model is reduced while minimizing the impact on what is learned (Reed, 1993). Naturally, a typical pruning algorithm starts *after* convergence to a minimum or training to some degree. This pretraining requirement has been left unattended until Lee et al. (2019) recently showed that pruning can be performed on untrained networks at initiailzation prior to training. They proposed a method called SNIP which relies on a new saliency criterion, namely *connection sensitivity*, defined as follows:

$$s_j(\mathbf{w}; \mathcal{D}) = \frac{|g_j(\mathbf{w}; \mathcal{D})|}{\sum_{k=1}^{m} |g_k(\mathbf{w}; \mathcal{D})|} \,, \qquad \text{where} \quad g_j(\mathbf{w}; \mathcal{D}) = \frac{\partial L(\mathbf{c} \odot \mathbf{w}; \mathcal{D})}{\partial c_j} \bigg|_{\mathbf{c}=\mathbf{1}} . \qquad (1)$$

Here, $s_j$ is the saliency of the parameter $j$, $\mathbf{w} \in \mathbb{R}^m$ is the network parameters, $\mathbf{c} \in \{0, 1\}^m$ is the auxiliary indicator variables representing the connectivity of network parameters, $m$ is the total number of parameters in the network, and $\mathcal{D}$ is a given dataset. Also, $g_j$ is the derivative of the loss $L$ with respect to $c_j$, which turns out to be an infinitesimal approximation of the change in the

---

[1] The term faithful is used to describe signals propagating in a network isometrically with minimal amplification or attenuation, and borrowed from Saxe et al. (2014), the first work to introduce dynamical isometry.

loss with respect to removing the parameter $j$. Designed to be computed at initialization, pruning is performed by keeping top-$\kappa$ (where $\kappa$ denotes a desired sparsity level) salient parameters based on the above sensitivity scores.

**Dynamical isometry and mean field theory**.  The success of training deep neural networks is due in large part to the initial weights (Hinton & Salakhutdinov, 2006; Glorot & Bengio, 2010; Pascanu et al., 2013).  In essence, the principle behind these random weight initializations is to have the mean squared singular value of a network's input-output Jacobian close to 1, so that on average, an error vector will preserve its norm under backpropagation; however, this is not sufficient to prevent amplification or attenuation of an error vector on worst case. A stronger condition that having as many singular values as possible near 1 is called *dynamical isometry* (Saxe et al., 2014). Under this condition, error signals backpropagate isometrically through the network, approximately preserving its norm and all angles between error vectors. Alongside dynamical isometry, *mean field theory* is used to develop a theoretical understanding of signal propagation in neural networks with random parameters (Poole et al., 2016). Precisely, the mean field approximation states that preactivations of wide, untrained neural networks can be captured as a Gaussian distribution. Recent works revealed a maximum depth through which signals can propagate at initialization, and verified that networks are trainable when signals can travel all the way through them (Schoenholz et al., 2017; Yang & Schoenholz, 2017; Xiao et al., 2018).

## 3 SIGNAL PROPAGATION PERSPECTIVE TO PRUNING RANDOM NETWORKS

**Problem setup**.  Consider a fully-connected, feed-forward neural network with weight matrices $\mathbf{W}^l \in \mathbb{R}^{N \times N}$, biases $\mathbf{b}^l \in \mathbb{R}^N$, pre-activations $\mathbf{h}^l \in \mathbb{R}^N$, and post-activations $\mathbf{x}^l \in \mathbb{R}^N$, for $l \in \{1 \ldots K\}$ up to $K$ layers. Now, the feed-forward dynamics of a network can be written as,

$$\mathbf{x}^l = \phi(\mathbf{h}^l) \,, \qquad \mathbf{h}^l = \mathbf{W}^l \mathbf{x}^{l-1} + \mathbf{b}^l \,, \qquad (2)$$

where $\phi : \mathbb{R} \to \mathbb{R}$ is an elementwise nonlinearity, and the input is denoted by $\mathbf{x}^0$. Given the network configuration, the parameters are initialized by sampling from a probability distribution, typically a zero mean Gaussian with scaled variance (LeCun et al., 1998; Glorot & Bengio, 2010).

### 3.1 EFFECT OF INITIALIZATION ON PRUNING

It is observed in Lee et al. (2019) that pruning results tend to improve when initial weights are drawn from a scaled Gaussian, or so-called variance scaling initialization (LeCun et al., 1998; Glorot & Bengio, 2010; He et al., 2015). As we wish to better understand the role of these random initial weights in pruning, we will examine the effect of varying initialization on the pruning results.

In essence, variance scaling schemes introduce normalization factors to adjust the variance $\sigma$ of the weight sampling distribution, which can be summarized as $\sigma \to \frac{\alpha}{\psi_l} \sigma$, where $\psi_l$ is a layerwise scalar that depends on an architecture specification such as the number of output neurons in the previous layer (*e.g.*, fan-in), and $\alpha$ is a global scalar throughout the network. Notice in case of a network with layers of the same width, the variance can be controlled by a single scalar $\gamma = \frac{\alpha}{\psi}$ as $\psi_l = \psi$ for all layers $l$. In particular, we take both linear and tanh multilayer perceptron networks (MLP) of layers $K = 7$ and width $N = 100$ on MNIST with $\sigma = 1$ as the default, similar to Saxe et al. (2014). We initialize these networks with different $\gamma$, compute the connection sensitivity, prune it, and then visualize layerwise the resulting sparsity patterns $\mathbf{c}$ as well as the corresponding connection sensitivity used for pruning in Figure 1.

It is seen in the sparsity patterns that for the tanh network, unlike the linear case, more parameters tend to be pruned in the later layers than the earlier layers. As a result, this limits the learning capability of the subnetwork critically when a high sparsity level is requested; *e.g.*, for $\bar{\kappa} = 90\%$, only a few parameters in later layers are retained after pruning. This is explained by the connection sensitivity plot. The sensitivity of parameters in the nonlinear network tends to decrease towards the later layers, and therefore, choosing the top-$\kappa$ parameters globally based on the sensitivity scores results in a subnetwork in which retained parameters are distributed highly non-uniformly and sparsely towards the end of the network. This result implies that the initial weights have a crucial effect on the connection sensitivity, and from there, the pruning results.

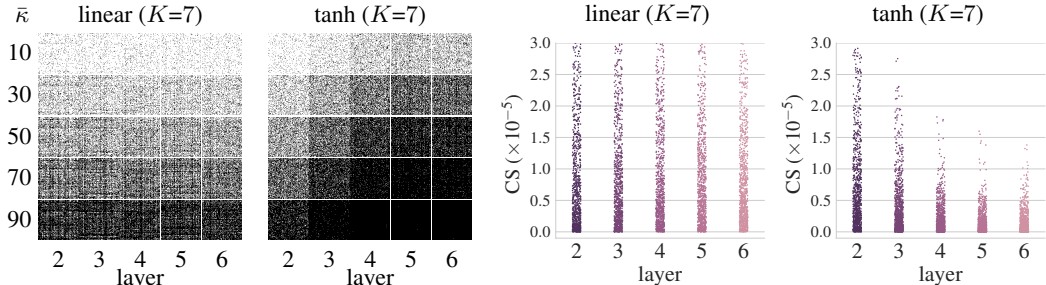

Figure 1: (left) layerwise sparsity patterns $c \in \{0, 1\}^{100 \times 100}$ obtained as a result of pruning for the sparsity level $\bar{\kappa} = \{10, .., 90\}\%$. Here, black(0)/white(1) pixels refer to pruned/retained parameters; (right) connection sensitivities (CS) measured for the parameters in each layer. All networks are initialized with $\gamma = 1.0$. Unlike the linear case, the sparsity pattern for the tanh network is non-uniform over different layers. When pruning for a high sparsity level (*e.g.*, $\bar{\kappa} = 90\%$), this becomes critical and leads to poor learning capability as there are only a few parameters left in later layers. This is explained by the connection sensitivity plot which shows that for the nonlinear network parameters in later layers have saturating, lower connection sensitivities than those in earlier layers.

## 3.2 GRADIENT SIGNAL IN CONNECTION SENSITIVITY

We posit that the unreliability of connection sensitivity observed in Figure 1 is due to poor signal propagation: an initialization that projects the input signal to be strongly amplified or attenuated in the forward pass will saturate the error signal under backpropagation (*i.e.*, gradients), and hence will result in poorly calibrated connection sensitivity scores across layers, which will eventually lead to poor pruning results, potentially with complete disconnection of signal paths (*e.g.*, entire layer).

Precisely, we give the relationship between the connection sensitivity and the gradients as follows. From Equation 1, connection sensitivity is a normalized magnitude of gradients with respect to the connectivity parameters $\mathbf{c}$. Here, we use the vectorized notation where $\mathbf{w}$ denotes all learnable parameters and $\mathbf{c}$ denotes the corresponding connectivity parameters. From chain rule, we can write:

$$\frac{\partial L(\mathbf{c} \odot \mathbf{w}; \mathcal{D})}{\partial \mathbf{c}}\bigg|_{\mathbf{c}=\mathbf{1}} = \frac{\partial L(\mathbf{c} \odot \mathbf{w}; \mathcal{D})}{\partial(\mathbf{c} \odot \mathbf{w})}\bigg|_{\mathbf{c}=\mathbf{1}} \odot \mathbf{w} = \frac{\partial L(\mathbf{w}; \mathcal{D})}{\partial \mathbf{w}} \odot \mathbf{w} \,. \tag{3}$$

Therefore, $\partial L/\partial \mathbf{c}$ is the gradients $\partial L/\partial \mathbf{w}$ amplified (or attenuated) by the corresponding weights $\mathbf{w}$, *i.e.*, $\partial L/\partial c_j = \partial L/\partial w_j \, w_j$ for all $j \in \{1 \ldots m\}$. Considering $\partial L/\partial c_j$ for a given $j$, since $w_j$ does not depend on any other layers or signal propagation, the only term that depends on signal propagation in the network is the gradient term $\partial L/\partial w_j$. Hence, a necessary condition to ensure faithful $\partial L/\partial \mathbf{c}$ (and connection sensitivity) is that the gradients $\partial L/\partial \mathbf{w}$ need to be faithful. In the following section, we formalize this from a signal propagation perspective, and characterize an initial condition that ensures reliable connection sensitivity measurement.

## 3.3 LAYERWISE DYNAMICAL ISOMETRY

### 3.3.1 GRADIENTS IN TERMS OF JACOBIANS

From the feed-forward dynamics of a network in Equation 2, the network's input-output Jacobian corresponding to a given input $\mathbf{x}^0$ can be written, by the chain rule of differentiation, as:

$$\mathbf{J}^{0,K} = \frac{\partial \mathbf{x}^K}{\partial \mathbf{x}^0} = \prod_{l=1}^{K} \mathbf{D}^l \mathbf{W}^l \,, \tag{4}$$

where $\mathbf{D}^l \in \mathbb{R}^{N \times N}$ is a diagonal matrix with entries $\mathbf{D}_{ij}^l = \phi'(h_i^l)\delta_{ij}$, with $\phi'$ denoting the derivative of nonlinearity $\phi$, and $\delta_{ij} = \mathbb{1}[i = j]$ is the Kronecker delta. Here, we will use $\mathbf{J}^{k,l}$ to denote the Jacobian from layer $k$ to layer $l$. Now, we give the relationship between gradients and Jacobians:

**Proposition 1.** Let $\epsilon = \partial L/\partial \mathbf{x}^K$ denote the error signal and $\mathbf{x}^0$ denote the input signal. Then,

1. the gradients satisfy:

$$\mathbf{g}_{\mathbf{w}^l}^T = \epsilon \, \mathbf{J}^{l,K} \mathbf{D}^l \otimes \mathbf{x}^{l-1} \, , \tag{5}$$

where $\mathbf{J}^{l,K} = \partial \mathbf{x}^K / \partial \mathbf{x}^l$ is the Jacobian from layer $l$ to the output and $\otimes$ is the Kronecker product.

2. additionally, for linear networks, *i.e.*, when $\phi$ is the identity:

$$\mathbf{g}_{\mathbf{w}^l}^T = \epsilon \, \mathbf{J}^{l,K} \otimes \left( \mathbf{J}^{0,l-1} \mathbf{x}^0 + \mathbf{a} \right) \, , \tag{6}$$

where $\mathbf{J}^{0,l-1} = \partial \mathbf{x}^{l-1} / \partial \mathbf{x}^0$ is the Jacobian from the input to layer $l-1$ and $\mathbf{a} \in \mathbb{R}^N$ is a constant term that does not depend on $\mathbf{x}^0$.

*Proof.* This can be proved by an algebraic manipulation of the chain rule while using the feedforward dynamics in Equation 2. We provide the full derivation in Appendix A. □

Notice that the gradient at layer $l$ constitutes both the backward propagation of the error signal $\epsilon$ up to layer $l$ and the forward propagation of the input signal $\mathbf{x}^0$ up to layer $l-1$. Moreover, especially in the linear case, the signal propagation in both directions is governed by the corresponding Jacobians. We believe that this interpretation of gradients is useful as it sheds light on how signal propagation affects the gradients. To this end, we next analyze the conditions on the Jacobians, which would guarantee faithful signal propagation in the network, and consequently, faithful gradients.

### 3.3.2 ENSURING FAITHFUL GRADIENTS

Here, we first consider the layerwise signal propagation which would be useful to derive properties on the initialization to ensure faithful gradients. To this end, let us consider the layerwise Jacobian:

$$\mathbf{J}^{l-1,l} = \frac{\partial \mathbf{x}^l}{\partial \mathbf{x}^{l-1}} = \mathbf{D}^l \mathbf{W}^l \, . \tag{7}$$

Note that it is sufficient to have *layerwise dynamical isometry* in order to ensure faithful signal propagation in the network.

**Definition 1.** (*Layerwise dynamical isometry*) Let $\mathbf{J}^{l-1,l} = \frac{\partial \mathbf{x}^l}{\partial \mathbf{x}^{l-1}} \in \mathbb{R}^{N_l \times N_{l-1}}$ be the Jacobian matrix of layer $l$. The network is said to satisfy layerwise dynamical isometry if the singular values of $\mathbf{J}^{l-1,l}$ are concentrated near 1 for all layers, *i.e.*, for a given $\epsilon > 0$, the singular value $\sigma_j$ satisfies $|1 - \sigma_j| \leq \epsilon$ for all $j$.

This would guarantee that the signal from layer $l$ to $l-1$ (or vice versa) is propagated without amplification or attenuation in any of its dimension. From Proposition 1 and Equation 7, by induction, it is easy to show that if the layerwise signal propagation is faithful, the error and input signals will faithfully propagate throughout the network, resulting in faithful gradients.

For linear networks, $\mathbf{J}^{l-1,l} = W^l$. Therefore, one can initialize the weight matrix to be *orthogonal* such that $(\mathbf{W}^l)^T \mathbf{W}^l = \mathbf{I}$, where $\mathbf{I}$ is the identity matrix of dimension $N$. In this case, all singular values of $\mathbf{W}^l$ are exactly 1 (*i.e.*, exact dynamical isometry), and such an initialization guarantees faithful gradients. While a linear network is of little practical use, we note that it helps to develop theoretical analysis and provides intuition as to why dynamical isometry is a useful measure.

For nonlinear networks, the diagonal matrix $\mathbf{D}^l$ needs to be accounted for as it depends on the pre-activations $\mathbf{h}^l$ at layer $l$. In this case, it is important to have the pre-activations $\mathbf{h}^l$ fall into the linear region of the nonlinear function $\phi$. Precisely, mean-field theory assumes that for large-$N$ limit, the empirical distribution of the pre-activations $\mathbf{h}^l$ converges to a Gaussian with zero mean and variance $q^l$, where the variance follows a recursion relation (Poole et al., 2016). Therefore, to achieve layerwise dynamical isometry, the idea becomes to find a fixed point $q^*$ such that $\mathbf{h}^l \sim \mathcal{N}(0, q^*)$ for all $l \in \{1 \ldots K\}$. Such a fixed point makes $\mathbf{D}^l = \mathbf{D}$ for all layers, and therefore, the pre-activations are placed in the linear region of the nonlinearity.[2] Then, given the nonlinearity, one can find a rescaling such that $(\mathbf{D}\mathbf{W}^l)^T(\mathbf{D}\mathbf{W}^l) = (\mathbf{W}^l)^T \mathbf{W}^l / \sigma_w^2 = \mathbf{I}$. The procedure for finding the rescaling $\sigma_w^2$ for various nonlinearities are discussed in Pennington et al. (2017; 2018). Also, this easily extends to convolutional neural networks using the initialization method in Xiao et al. (2018).

---

[2] Dynamical isometry can hold for antisymmetric sigmoidal activation functions (*e.g.*, tanh) as shown in Pennington et al. (2017). A recent work by Tarnowski et al. (2019) have also shown that dynamical isometry is achievable irrespective of the activation function in ResNets.

Table 1: Jacobian singular values and resulting sparse networks for the 7-layer tanh MLP network considered in section 3.1. SG, CN, and Sparsity refer to Scaled Gaussian, Condition Number (*i.e.*, $s_{max}/s_{min}$, where $s_{max}$ and $s_{min}$ are the maximum and minimum Jacobian singular values), and a ratio of pruned prameters to the total number of parameters, respectively. SG ($\gamma=10^{-2}$) is equivalent to the variance scaling initialization as in LeCun et al. (1998); Glorot & Bengio (2010). The failure cases correspond to unreliable connection sensitivity resulted from poorly conditioned initial Jacobians.

| | Jacobian singular values | | | Sparsity in pruned network (across layers) | | | | | | | |
|---|---|---|---|---|---|---|---|---|---|---|---|
| Initialization | Mean | Std | CN | 1 | 2 | 3 | 4 | 5 | 6 | 7 | Error |
| SG ($\gamma=10^{-4}$) | 2.46e−07 | 9.90e−08 | 4.66e+00 | 0.97 | 0.80 | 0.80 | 0.80 | 0.80 | 0.81 | 0.48 | 2.66 |
| SG ($\gamma=10^{-3}$) | 5.74e−04 | 2.45e−04 | 8.54e+00 | 0.97 | 0.80 | 0.80 | 0.80 | 0.80 | 0.81 | 0.48 | 2.67 |
| SG ($\gamma=10^{-2}$) | 4.49e−01 | 2.51e−01 | 5.14e+01 | 0.96 | 0.80 | 0.80 | 0.80 | 0.81 | 0.81 | 0.49 | 2.67 |
| SG ($\gamma=10^{-1}$) | 2.30e+01 | 2.56e+01 | 2.92e+04 | 0.96 | 0.81 | 0.82 | 0.82 | 0.82 | 0.80 | 0.45 | 2.61 |
| SG ($\gamma=10^{0}$) | 1.03e+03 | 2.61e+03 | 3.34e+11 | 0.85 | 0.88 | 0.99 | 1.00 | 1.00 | 1.00 | 0.91 | 90.2 |
| SG ($\gamma=10^{1}$) | 3.67e+04 | 2.64e+05 | inf | 0.84 | 0.95 | 1.00 | 1.00 | 1.00 | 1.00 | 1.00 | 90.2 |

We note that dynamical isometry is in fact a weaker condition than layerwise dynamical isometry. However, in practice, the initialization suggested in the existing works (Pennington et al., 2017; Xiao et al., 2018), *i.e.*, orthogonal initialization for weight matrices in each layer with rescaling based on mean field theory, satisfy layerwise dynamical isometry, even though this term was not mentioned.

Now, recall from Section 3.1 that a network is pruned with a global threshold based on connection sensitivity, and from Section 3.2 that the connection sensitivity is the gradients scaled by the weights. This in turn implies that the connection sensitivity scores across layers are required to be of the same scale. To this end, we require the gradients to be faithful and the weights to be in the same scale for all the layers. Notice, this condition is trivially satisfied when the layerwise dynamical isometry is ensured, as each layer is initialized identically (*i.e.*, orthogonal initialization) and the gradients are guaranteed to be faithful.

Finally, we verify the failure of pruning cases presented in Section 3.1 based on the signal propagation perspective. Specifically, we measure the singular value distribution of the input-output Jacobian ($\mathbf{J}^{0,K}$) for the 7-layer tanh MLP network, and the results are reported in Table 1. Note that while connection sensitivity based pruning is robust to moderate changes in the Jacobian singular values, it failed catastrophically when the condition number of the Jacobian is very large ($> 1e+11$). In fact, these failure cases correspond to the completely disconnected networks, as a consequence of pruning with unreliable connection sensitivity resulted from poorly conditioned initial Jacobians. As we will show subsequently, these findings extend to modern architectures, and layerwise dynamical isometry yields well-conditioned Jacobians and in turn the best pruning results.

## 4 SIGNAL PROPAGATION IN SPARSE NEURAL NETWORKS

So far, we have shown empirically and theoretically that layerwise dynamical isometry can improve the process of pruning at initialization. One remaining question to address is the following: how well do signals propagate in the pruned sparse networks? In this section, we first examine the effect of sparsity on signal propagation after pruning. We find that indeed pruning can break dynamical isometry, degrading trainability of sparse networks. Then we follow up to present a simple, but effective data-free method to recover approximate dynamical isometry on sparse networks.

**Setup**. The overall process is summarized as follows: Step 1. Initialize a network with a variance scaling (VS) or layerwise dynamical isometry (LDI) satisfying orthogonal initialization. Step 2. Prune at initialization for a sparsity level $\bar{\kappa}$ based on connection sensitivity (CS); we also test random (Rand) and magnitude (Mag) based pruning for comparison. Step 3. (optional) Enforce approximate dynamical isometry, if specified. Step 4. Train the pruned sparse network using SGD. We measure signal propagation (*e.g.*, Jacobian singular values) on the sparse network right before Step 4, and observe training behavior during Step 4. Different methods are named as {A}-{B}-{C}, where A, B, C stand for initialization scheme, pruning method, (optional) approximate isometry, respectively. We perform this on 7-layer linear and tanh MLP networks as before [3].

---

[3] We conduct the same experiments for ReLU and Leaky-ReLU activation functions (see Appendix C).

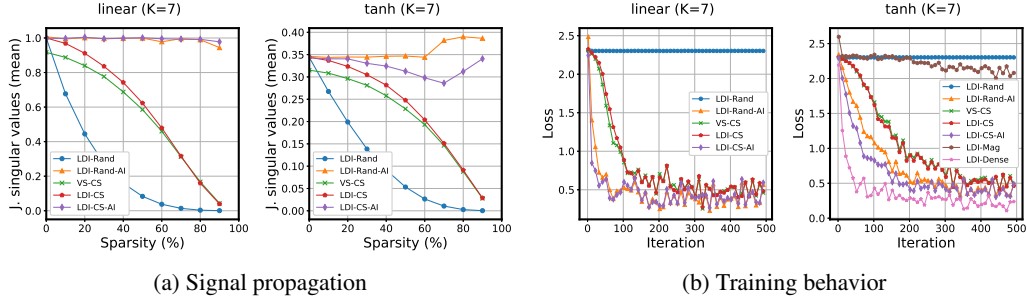

(a) Signal propagation              (b) Training behavior

Figure 2: (a) Signal propagation (mean Jacobian singular values) in sparse networks pruned for varying sparsity levels $\bar{\kappa}$, and (b) training behavior of the sparse network at $\bar{\kappa} = 90\%$. Signal propagation, pruning scheme, and overparameterization affect trainability of sparse neural networks. We train using SGD with the initial learning rate of 0.1 decayed by 1/10 at every 20k iterations. All results are the average over 10 runs. We provide other singular value statistics (max, min, std), accuracy plot, and extended training results for random and magnitude pruning in Appendix C.

**Effect of pruning on signal propagation and trainability**. Let us first check signal propagation measurements in the pruned networks (see Figure 2a). In general, Jacobian singular values decrease continuously as the sparsity level increases (except for $\{\cdot\}$-$\{\cdot\}$-AI which we will explain later), indicating that the more parameters are removed, the less faithful a network is likely to be with regard to propagating signals. Also, notice that the singular values drop more rapidly with random pruning compared to connection sensitivity based pruning methods (*i.e.*, $\{\cdot\}$-Rand vs. $\{\cdot\}$-CS). This means that pruning using connection sensitivity is more robust to destruction of dynamical isometry and preserve better signal propagation in the sparse network than random pruning. We further note that, albeit marginal, layerwise dynamical isometry allows better signal propagation than variance scaling initialization, with relatively higher mean singular values and much lower standard deviations especially in the low sparsity regime (see Appendix C).

Now, we look into the relation between signal propagation and trainability of the sparse networks. Figure 2b shows training behavior of the pruned networks ($\bar{\kappa} = 90\%$) obtained by different methods. We can see a clear correlation between signal propagation capability of a network and its training performance; *i.e.*, the better a network propagates signals, the faster it converges during training. For instance, compare the trainability of a network before and after pruning. That is, compared to LDI-Dense ($\bar{\kappa} = 0$), LDI-$\{$CS, Mag, Rand$\}$ decrease the loss much slowly; random pruning starts to decrease the loss around 4k iteration, and finally reaches to close to zero loss around 10k iterations (see Appendix C), which is more than an order of magnitude slower than a network pruned by connection sensitivity. Recall that the pruned networks have much smaller singular values.

**Enforcing approximate dynamical isometry**. The observation above indicates that the better signal propagation is ensured on sparse networks, the better their training performs. This motivates us to think of the following: what if we can *repair* the broken isometry, before we start training the pruned network, such that we can achieve trainability comparable to that of the dense network? Precisely, we consider the following:

$$\min_{\mathbf{W}^l} \|(\mathbf{C}^l \odot \mathbf{W}^l)^T (\mathbf{C}^l \odot \mathbf{W}^l) - \mathbf{I}^l\|_F, \tag{8}$$

where $\mathbf{C}^l, \mathbf{W}^l, \mathbf{I}^l$ are the sparse mask obtained by pruning, the corresponding weights, the identity matrix at layer $l$, respectively, and $\|\cdot\|_F$ is the Frobenius norm. We optimize this for all layers identically using gradient descent. Given the sparsity topology $\mathbf{C}^l$ and initial weights $\mathbf{W}^l$, this data-free method attempts to find an optimal $\mathbf{W}^*$ such that the combination of the sparse topology and the weights to be layerwise orthogonal, potentially to the full rank capacity. This simple method (*i.e.*, $\{\cdot\}$-$\{\cdot\}$-AI, where AI is named for Approximate Isometry) turns out to be highly effective. The results are provided in Figure 2, and we summarize our key findings below:

- *Signal propagation* (LDI-$\{$CS, Rand$\}$ vs. LDI-$\{$CS, Rand$\}$-AI). The decreased singular values (by pruning $\bar{\kappa} > 0$) bounce up dramatically and become close to the level before pruning. This means that orthogonality enforced by Equation 8 is achieved in the sparse topology of the pruned

Table 2: Pruning results for various neural networks on different datasets. All networks are pruned at initialization for the sparsity $\bar{\kappa} = 90\%$ based on connection sensitivity scores as in Lee et al. (2019). We report orthogonality scores (OS) and generalization errors (Error) on CIFAR-10 (VGG16, ResNets) and Tiny-ImageNet (WRN16); all results are the average over 5 runs. The **first** and second best results are highlighted in each column of errors. The orthogonal initialization with enforced approximate isometry method (*i.e.*, LDI-AI) achieves the best results across all tested architectures.

| | VGG16 | | ResNet32 | | ResNet56 | | ResNet110 | | WRN16 | |
|---|---|---|---|---|---|---|---|---|---|---|
| Initialization | OS | Error | OS | Error | OS | Error | OS | Error | OS | Error |
| VS-L | 13.72 | 8.16 | 4.50 | 11.96 | 4.64 | 10.43 | 4.65 | 9.13 | 11.99 | 45.08 |
| VS-G | 13.60 | 8.18 | 4.55 | 11.89 | 4.67 | 10.60 | 4.67 | 9.17 | 11.50 | 44.56 |
| VS-H | 15.44 | 8.36 | 4.41 | 12.21 | 4.44 | 10.63 | 4.39 | 9.08 | 13.49 | 46.62 |
| LDI | 13.33 | 8.11 | 4.43 | 11.55 | 4.51 | 10.08 | 4.57 | 8.88 | 11.28 | 44.20 |
| LDI-AI | 6.43 | **7.99** | 2.62 | **11.47** | 2.79 | **9.85** | 2.92 | **8.78** | 6.62 | **44.12** |

Table 3: Pruning results for VGG16 and ResNet32 with different activation functions on CIFAR-10. We report generalization errors (avg. over 5 runs), and the **first** and second best results are highlighted.

| | VGG16 | | | ResNet32 | | |
|---|---|---|---|---|---|---|
| Initialization | tanh | l-relu | selu | tanh | l-relu | selu |
| VS-L | 9.07 | 7.78 | 8.70 | 13.41 | 12.04 | 12.26 |
| VS-G | 9.06 | 7.84 | 8.82 | 13.44 | 12.02 | 12.32 |
| VS-H | 9.99 | 8.43 | 9.09 | **13.12** | 11.66 | 12.21 |
| LDI | 8.76 | 7.53 | 8.21 | 13.22 | 11.58 | 11.98 |
| LDI-AI | **8.72** | **7.47** | **8.20** | 13.14 | **11.51** | **11.68** |

Table 4: Unsupervised pruning results for $K$-layer MLP networks on MNIST. All networks are pruned for sparsity $\bar{\kappa} = 90\%$ at orthogonal initialization. We report generalization errors (avg. over 10 runs).

| Loss | Superv. | K=3 | K=5 | K=7 |
|---|---|---|---|---|
| GT | ✓ | 2.46 | 2.43 | 2.61 |
| Pred. (raw) | ✗ | 3.31 | 3.38 | 3.60 |
| Pred. (softmax) | ✗ | 3.11 | 3.37 | 3.56 |
| Unif. | ✗ | 2.77 | 2.77 | 2.94 |

network (*i.e.*, *approximate dynamical isometry*), and therefore, signal propagation on the sparse network is likely to behave similarly to the dense network. As expected, the training performance increased significantly (*e.g.*, compare LDI-CS with LDI-CS-AI for trainability). This works more dramatically for random pruning; *i.e.*, even for randomly pruned sparse networks, training speed increases significantly, implying the benefit of ensuring signal propagation.

- *Structure* (LDI-Rand-AI vs. LDI-CS-AI). Even if the approximate dynamical isometry is enforced identically, the network pruned using connection sensitivity shows better trainability than the randomly pruned network. This potentially means that the sparse topology obtained by different pruning methods also matters, in addition to signal propagation characteristics.
- *Overparameterization* (LDI-Dense vs. LDI-{CS, Rand}-AI). Even though the singular values are restored to a level close to before pruning with approximate isometry, the non-pruned dense network converges faster than pruned networks. We hypothesize that in addition to signal propagation, overparameterization helps in optimization taking less time to find a minimum.

While being simple and data free (thus fast), our signal propagation perspective not only can be used to improve trainability of sparse neural networks, but also to complement a common explanation for decreased trainability of compressed networks which is often attributed merely to a reduced capacity. Our results also extend to the case of convolutional neural network (see Figure 8 in Appendix C).

## 5 VALIDATION AND EXTENSIONS

In this section, we aim to demonstrate the efficacy of our signal propagation perspective on a wide variety of settings. We first evaluate the idea of employing layerwise dynamical isometry on various modern neural networks. In addition, we further study the role of supervision under the pruning at initialization regime, extending it to unsupervised pruning. Our results show that indeed, pruning can be approached from the signal propagation perspective at varying scale, bringing forth the notion of neural architecture sculpting. The experiment settings used to generate the presented results are detailed in Appendix B. The code can be found here: https://github.com/namhoonlee/spp-public.

## 5.1 EVALUATION ON VARIOUS NEURAL NETWORKS AND DATASETS

Here, we verify that our signal propagation perspective for pruning neural networks at initialization is indeed valid, by evaluating further on various modern neural networks and datasets. To this end, we provide orthogonality scores (OS) and generalization errors of the sparse networks obtained by different methods and show that layerwise dynamical isometry with enforced approximate isometry results in the best performance; here, we define OS as $\frac{1}{l} \sum_l \|(\mathbf{C}^l \odot \mathbf{W}^l)^T (\mathbf{C}^l \odot \mathbf{W}^l) - \mathbf{I}^l\|_F$, which can be used to indicate how close are the weight matrices in each layer of the pruned network to being orthogonal. All results are the average of 5 runs, and we do not optimize anything specific for a particular case (see Appendix B for experiment settings). The results are presented in Table 2.

The **best** pruning results are achieved when the approximate dynamical isometry is enforced on the pruned sparse network (*i.e.*, LDI-AI), across all tested architectures. Also, the second best results are achieved with the orthogonal initialization that satisfies layerwise dynamical isometry (*i.e.*, LDI). Looking closely, it is evident that there exists a high correlation between the orthogonality scores and the performance of pruned networks; *i.e.*, the network initialized to have the lowest orthogonality scores achieves the best generalization errors after training. Note that the orthogonality scores being close to 0, by definition, states how faithful a network will be with regard to letting signals propagate without being amplified or attenuated. Therefore, the fact that a pruned network with the lowest orthogonality scores tends to yield good generalization errors further validates that our signal propagation perspective is indeed effective for pruning at initialization. Moreover, we test for other nonlinear activation functions (tanh, leaky-relu, selu), and found that the orthogonal initialization consistently outperforms variance scaling methods (see Table 3).

## 5.2 PRUNING WITHOUT SUPERVISION

So far, we have shown that pruning random networks can be approached from a signal propagation perspective by ensuring faithful connection sensitivity. Another factor that constitutes connection sensitivity is the loss term. At a glance, it is not obvious how informative the supervised loss measured on a random network will be for connection sensitivity. In this section, we look into the effect of supervision, by simply replacing the loss computed using ground-truth labels with different unsupervised surrogate losses as follows: replacing the target distribution using ground-truth labels with uniform distribution (Unif.), and using the averaged output prediction of the network (Pred.; softmax/raw). The results for MLP networks are in Table 4. Even though unsupervised pruning results are not as good as the supervised case, the results are still interesting, especially for the uniform case, in that there was no supervision given. We thus experiment further for the uniform case on other networks, and obtain the following results: 8.25, 11.69, 11.01, 8.82 errors (%) for VGG16, ResNet32, ResNet56, ResNet110, respectively. Surprisingly, the results are often competitive to that of pruning with supervision (*i.e.*, compare to LDI results in Table 2). Notably, previous pruning algorithms assume the existence of supervision a priori. Being the first demonstration, along with the signal propagation perspective, this unsupervised pruning strategy can be useful in scenarios where there are no labels or only weak supervision is available.

To demonstrate further, we also conducted *transfer of sparsity* experiments such as transferring a pruned network from one task to another (MNIST $\leftrightarrow$ Fashion-MNIST). Table 5 shows that, while pruning results may degrade if sparsity is transferred, or done without supervision, less impact is caused for unsupervised pruning when transferred to a different task (*i.e.*, 0.52 to 0.14 on MNIST, and 1.11 to −0.78 on F-

Table 5: Transfer of sparsity experiment results for LeNet. We prune for $\bar{\kappa} = 97\%$ at orthogonal initialization, and report gen. errors (average over 10 runs).

| | Dataset | | Error | | | Error |
|---|---|---|---|---|---|---|
| **Category** | prune | train&test | sup. $\rightarrow$ unsup. | ($\Delta$) | | rand |
| Standard | MNIST | MNIST | $2.42 \rightarrow 2.94$ | +0.52 | | 15.56 |
| Transfer | F-MNIST | MNIST | $2.66 \rightarrow 2.80$ | **+0.14** | | 18.03 |
| Standard | F-MNIST | F-MNIST | $11.90 \rightarrow 13.01$ | +1.11 | | 24.72 |
| Transfer | MNIST | F-MNIST | $14.17 \rightarrow 13.39$ | **-0.78** | | 24.89 |

MNIST). This indicates that inductive bias exists in data, affecting transfer and unsupervised pruning, and potentially, that "universal" sparse topology might be obtainable if universal data distribution is known (*e.g.*, extremely large dataset in practice). This may help in situations where different tasks from unknown data distribution are to be performed (*e.g.*, continual learning). We also tested two other unsupervised losses, but none performed as well as uniform loss (*e.g.*, Jacobian norms $\|J\|_1$: 5.03, $\|J\|_2$: 3.00 vs. Unif.: 2.94), implying that if pruning is to be unsupervised, the uniform loss would better be used, because other unsupervised losses depend on the input data (thus can suffer from inductive bias). Random pruning degrades significantly at high sparsity for all cases.

### 5.3 NEURAL ARCHITECTURE SCULPTING

We have shown that pruning at initialization, even when no supervision is provided, can be effective based on the signal propagation perspective. This begs the question of whether pruning needs to be limited to pre-shaped architectures or not. In other words, what if pruning is applied to an arbitrarily bulky network and is treated as *sculpting* an architecture? In order to find out, we conduct the following experiments: we take a popular pre-designed architecture (ResNet20 in He et al. (2016)) as a base network, and consider a range of variants that are originally bigger than the base model, but pruned to have the same number of parameters as the base dense network. Specifically, we consider the following equivalents: **(1)** the same number of residual blocks, but with larger widths; **(2)** a reduced number of residual blocks with larger widths; **(3)** a larger residual block and the same width (see Table 6 in Appendix B for details). The results are presented in Figure 3.

Overall, the sparse equivalents record lower errors than the dense base model. Notice that some models are extremely sparse (*e.g.*, Equivalent 1 pruned for $\bar{\kappa} = 98.4\%$). While all networks have the same number of parameters, discovered sparse equivalents outperform the dense reference network. This result is well aligned with recent findings in Kalchbrenner et al. (2018): large sparse networks can outperform their small dense counterpart, while enjoying increased computational and memory efficiency via a dedicated implementation for sparsity in practice. Also, it seems that pruning wider networks tends to be more effective in producing a better model than pruning deeper ones (*e.g.*, Equivalent 1 vs. Equivalent 3). We further note that unlike existing prior works, the sparse networks are discovered by sculpting arbitrarily-designed architecture, without pretraining nor supervision.

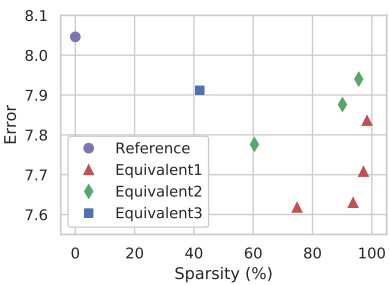

Figure 3: Neural architecture sculpting results on CIFAR-10. We report generalization errors (avg. over 5 runs). All networks have the same number of parameters (269k) and trained identically.

## 6 DISCUSSION AND FUTURE WORK

In this work, we have approached the problem of pruning neural networks at initialization from a signal propagation perspective. Based on observing the effect of varying the initialization, we found that initial weights have a critical impact on connection sensitivity measurements and hence pruning results. This led us to conduct theoretical analysis based on dynamical isometry and a mean field theory, and formally characterize a sufficient condition to ensure faithful signal propagation in a given network. Moreover, our analysis on compressed neural networks revealed that signal propagation characteristics of a sparse network highly correlates with its trainability, and also that pruning can break dynamical isometry ensured on a network at initialization, resulting in degradation of trainability of the compressed network. To address this, we introduced a simple, yet effective data-free method to recover the orthogonality and enhance trainability of the compressed network. Finally, throughout a range of validation and extension experiments, we verified that our signal propagation perspective is effective for understanding, improving, and extending the task of pruning at initialization across various settings. We believe that our results on the increased trainability of compressed networks can take us one step towards finding "winning lottery ticket" (*i.e.*, a set of initial weights that given a sparse topology can quickly reach to a generalization performance that is comparable to the uncompressed network, once trained) suggested in Frankle & Carbin (2019).

We point out, however, that there remains several aspects to consider. While pruning on enforced isometry produces trainable sparse networks, the two-stage orthogonalization process (*i.e.*, prune first and enforce the orthogonality later) can be suboptimal especially at a high sparsity level. Also, network weights change during training, which can affect signal propagation characteristics, and therefore, dynamical isometry may not continue to hold over the course of training. We hypothesize that a potential key to successful neural network compression is to address the complex interplay between optimization and signal propagation, and it might be immensely beneficial if an optimization naturally takes place in the space of isometry. We believe that our signal propagation perspective provides a means to formulate this as an optimization problem by maximizing the trainability of sparse networks while pruning, and we intend to explore this direction as a future work.

ACKNOWLEDGMENTS

This work was supported by the ERC grant ERC-2012-AdG 321162-HELIOS, EPSRC grant See-bibyte EP/M013774/1, EPSRC/MURI grant EP/N019474/1 and the Australian Research Council Centre of Excellence for Robotic Vision (project number CE140100016). We would also like to acknowledge the Royal Academy of Engineering and FiveAI, and thank Richard Hartley, Puneet Dokania and Amartya Sanyal for helpful discussions.

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

## A    GRADIENTS IN TERMS OF JACOBIANS

**Proposition 1.** Let $\epsilon = \partial L/\partial \mathbf{x}^K$ denote the error signal and $\mathbf{x}^0$ denote the input signal. Then,

1. the gradients satisfy:
$$\mathbf{g}_{\mathbf{w}^l}^T = \epsilon \mathbf{J}^{l,K} \mathbf{D}^l \otimes \mathbf{x}^{l-1} \,, \tag{9}$$
where $\mathbf{J}^{l,K} = \partial \mathbf{x}^K/\partial \mathbf{x}^l$ is the Jacobian from layer $l$ to the output and $\otimes$ is the Kronecker product.
2. additionally, for linear networks, *i.e.*, when $\phi$ is the identity:
$$\mathbf{g}_{\mathbf{w}^l}^T = \epsilon \mathbf{J}^{l,K} \otimes \left(\mathbf{J}^{0,l-1}\mathbf{x}^0 + \mathbf{a}\right) \,, \tag{10}$$
where $\mathbf{J}^{0,l-1} = \partial \mathbf{x}^{l-1}/\partial \mathbf{x}^0$ is the Jacobian from the input to layer $l-1$ and $\mathbf{a} \in \mathbb{R}^N$ is the constant term that does not depend on $\mathbf{x}^0$.

*Proof.* The proof is based on a simple algebraic manipulation of the chain rule. The gradient of the loss with respect to the weight matrix $\mathbf{W}^l$ can be written as:
$$\mathbf{g}_{\mathbf{w}^l} = \frac{\partial L}{\partial \mathbf{W}^l} = \frac{\partial L}{\partial \mathbf{x}^K}\frac{\partial \mathbf{x}^K}{\partial \mathbf{x}^l}\frac{\partial \mathbf{x}^l}{\partial \mathbf{W}^l} \,. \tag{11}$$
Here, the gradient $\partial \mathbf{y}/\partial \mathbf{x}$ is represented as a matrix of dimension $\mathbf{y}$-size $\times$ $\mathbf{x}$-size. For gradients with respect to matrices, their vectorized from is used. Notice,
$$\frac{\partial \mathbf{x}^l}{\partial \mathbf{W}^l} = \frac{\partial \mathbf{x}^l}{\partial \mathbf{h}^l}\frac{\partial \mathbf{h}^l}{\partial \mathbf{W}^l} = \mathbf{D}^l\frac{\partial \mathbf{h}^l}{\partial \mathbf{W}^l} \,. \tag{12}$$
Considering the feed-forward dynamics for a particular neuron $i$,
$$h_i^l = \sum_j W_{ij}^l x_j^{l-1} + b_i^l \,, \tag{13}$$
$$\frac{\partial h_i^l}{\partial W_{ij}^l} = x_j^{l-1} \,.$$
Therefore, using the Kronecker product, we can compactly write:
$$\frac{\partial \mathbf{x}^l}{\partial \mathbf{W}^l} = (\mathbf{D}^l)^T \otimes (\mathbf{x}^{l-1})^T \,. \tag{14}$$
Now, Equation 11 can be written as:
$$\mathbf{g}_{\mathbf{w}^l} = (\epsilon \mathbf{J}^{l,K}\mathbf{D}^l)^T \otimes (\mathbf{x}^{l-1})^T \,, \tag{15}$$
$$\mathbf{g}_{\mathbf{w}^l}^T = \epsilon \mathbf{J}^{l,K}\mathbf{D}^l \otimes \mathbf{x}^{l-1} \,.$$
Here, $\mathbf{A}^T \otimes \mathbf{B}^T = (\mathbf{A} \otimes \mathbf{B})^T$ is used. Moreover, for linear networks $\mathbf{D}^l = \mathbf{I}$ and $\mathbf{x}^l = \mathbf{h}^l$ for all $l \in \{1 \ldots K\}$. Therefore, $\mathbf{x}^{l-1}$ can be written as:
$$\begin{aligned}\mathbf{x}^{l-1} &= \phi(\mathbf{W}^{l-1}\phi(\mathbf{W}^{l-2}\ldots\phi(\mathbf{W}^1\mathbf{x}^0 + \mathbf{b}^1)\ldots + \mathbf{b}^{l-2}) + \mathbf{b}^{l-1}) \,, \tag{16}\\ &= \mathbf{W}^{l-1}(\mathbf{W}^{l-2}\ldots(\mathbf{W}^1\mathbf{x}^0 + \mathbf{b}^1)\ldots + \mathbf{b}^{l-2}) + \mathbf{b}^{l-1} \,, \\ &= \prod_{k=1}^{l-1}\mathbf{W}^k\mathbf{x}^0 + \prod_{k=2}^{l-1}\mathbf{W}^k\mathbf{b}^1 + \ldots + \mathbf{b}^{l-1} \,, \\ &= \mathbf{J}^{0,l-1}\mathbf{x}^0 + \mathbf{a} \,,\end{aligned}$$
where $\mathbf{a}$ is the constant term that does not depend on $\mathbf{x}^0$. Hence, the proof is complete. $\square$

# B  EXPERIMENT SETTINGS

**Pruning at initialization**.  By default, we perform pruning at initialization based on connection sensitivity scores as in Lee et al. (2019). When computing connection sensitivity, we always use all examples in the training set to prevent stochasticity by a particular mini-batch. Unless stated otherwise, we set the default sparsity level to be $\bar{\kappa} = 90\%$ (*i.e.*, 90% of the entire parameters in a network is pruned away). For all tested architectures, pruning for such level of sparsity does not lead to a large accuracy drop. Additionally, we perform either random pruning (at initialization) or a magnitude based pruning (at pretrained) for comparison purposes. Random pruning refers to pruning parameters randomly and globally for a given sparsity level. For the magnitude based pruning, we first train a model and simply prune parameters globally in a single-shot based on the magnitude of the pretrained parameters (*i.e.*, keep the large weights while pruning small ones). For initialization methods, we follow either variance scaling initialization schemes (*i.e.*, VS-L, VS-G, VS-H, as in LeCun et al. (1998); Glorot & Bengio (2010); He et al. (2015), respectively) or (convolutional) orthogonal initialization schemes (Saxe et al., 2014; Xiao et al., 2018).

**Training and evaluation**.  Throughout experiments, we evaluate pruning results on MNIST, CIFAR-10, and Tiny-ImageNet image classification tasks. For training of the pruned sparse networks, we use SGD with momentum and train up to 80k (for MNIST) or 100k (for CIFAR-10 and Tiny-ImageNet) iterations. The initial learning rate is set to be 0.1 and is decayed by 1/10 at every 20k (MNIST) or 25k (CIFAR-10 and Tiny-ImageNet). The mini-batch size is set to be 100, 128, 200 for MNIST, CIFAR-10, Tiny-ImageNet, respectively. We do not optimize anything specific for a particular case, and follow the standard training procedure. For all experiments, we use 10% of training set for the validation set, which corresponds to 5400, 5000, 9000 images for MNIST, CIFAR-10, Tiny-IamgeNet, respectively. We evaluate at every 1k iteration, and record the lowest test error. All results are the average of either 10 (for MNIST) or 5 (for CIFAR-10 and Tiny-ImageNet) runs.

**Signal propagation and approximate dynamical isometry**.  We use the entire training set when computing Jacobian singular values of a network. In order to enforce approximate dynamical isometry when specified, given a pruned sparse network, we optimize for the objective in Equation 8 using gradient descent. The learning rate is set to be 0.1 and we perform 10k gradient update steps (although it usually reaches to convergence far before). This process is data-free and thus fast; *e.g.*, depending on the size of the network and the number of update steps, it can take less than a few seconds on a modern computer.

**Neural architecture sculpting**.  We provide the model details in Table 6.

Table 6: All models (Equivalents 1,2,3) are initially bigger than the base network (ResNet20), by either being wider or deeper, but pruned to have the same number of parameters as the base network (269k). The widening factor (k) refers to the filter multiplier; *e.g.*, for the basic filter size of 16, the widening factor of k=2 will result in 32 filters. The block size refers to the number of residual blocks in each block layer; all models have three block layers. More/less number of residual blocks means the network is deeper/shallower. The reported generalization errors are averages over 5 runs. We find that the technique of *architecture sculpting*, pruning randomly initialized neural networks based on our signal propagation perspective even in the absence of ground-truth supervision, can be used to find models of superior performance under the same parameter budget.

| Model category | Shape | Widening (k) | Block size | Init. | GT | Sparsity | Error |
|---|---|---|---|---|---|---|---|
| Base | ResNet20 (He et al., 2016) | 1 | 3 | VS-H | ✓ | 0.0 | 8.046 |
| Equivalent 1 | wider | 2 | 3 | LDI | ✗ | 74.8 | 7.618 |
| | wider | 4 | 3 | LDI | ✗ | 93.7 | 7.630 |
| | wider | 6 | 3 | LDI | ✗ | 97.2 | 7.708 |
| | wider | 8 | 3 | LDI | ✗ | 98.4 | 7.836 |
| Equivalent 2 | wider & shallower | 2 | 2 | LDI | ✗ | 60.4 | 7.776 |
| | wider & shallower | 4 | 2 | LDI | ✗ | 90.1 | 7.876 |
| | wider & shallower | 6 | 2 | LDI | ✗ | 95.6 | 7.940 |
| Equivalent 3 | deeper | 1 | 5 | LDI | ✗ | 42.0 | 7.912 |

# C  SIGNAL PROPAGATION IN SPARSE NETWORKS: ADDITIONAL RESULTS

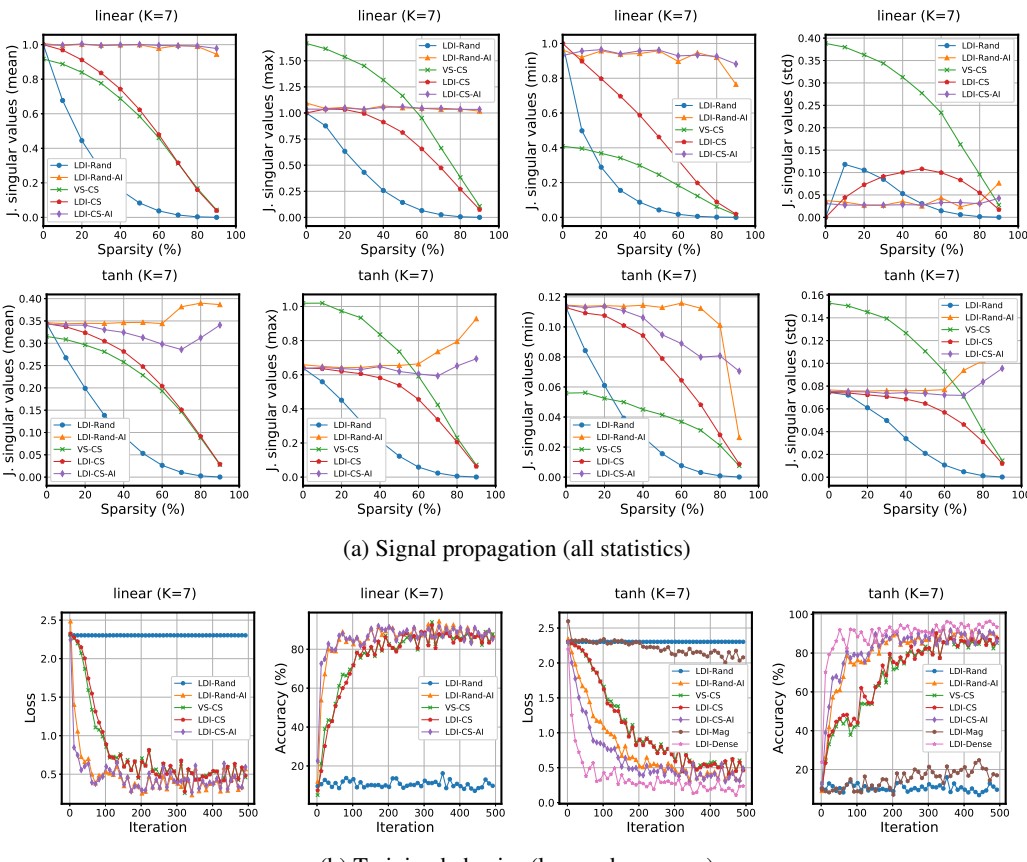

(a) Signal propagation (all statistics)

(b) Training behavior (loss and accuracy)

Figure 4: Full results for (a) signal propagation (all singular value statistics), and (b) training behavior (including accuracy) for 7-layer linear and tanh MLP networks. We provide results of LDI-Rand, LDI-Rand-AI, VS-CS, LDI-CS, LDI-CS-AI on the linear case for both singular value statistics and training log. We also plot results of LDI-Mag and LDI-Dense on the tanh case for trainability; the training results of non-pruned (LDI-Dense) and magnitude (LDI-Mag) pruning are only reported for the tanh case, because the learning rate had to be lowered for the linear case (otherwise it explodes), which makes the comparison not entirely fair. We provide the singular value statistics for the magnitude pruning in Figure 6 to avoid clutter. Also, extended training logs for random and magnitude based pruning are provided separately in Figure 5 to illustrate the difference in convergence speed.

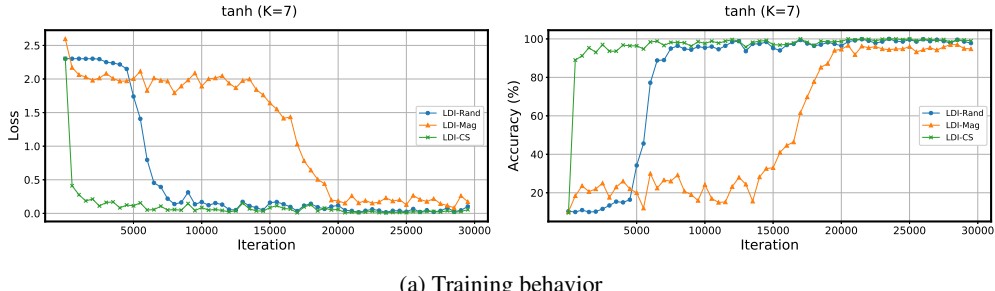

(a) Training behavior

Figure 5: Extended training log (*i.e.*, Loss and Accuracy) for random (Rand) and magnitude (Mag) pruning. The sparse networks obtained by random or magnitude pruning take a much longer time to train than that obtained by pruning based on connection sensitivity. All methods are pruned at the layerwise orthogonal initialization, and trained the same way as before.

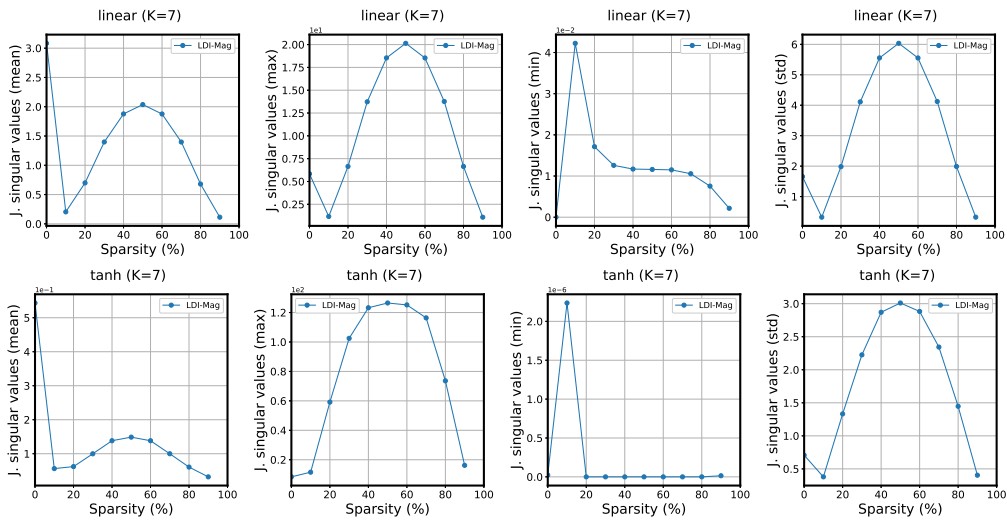

(a) Signal propagation (all statistics; magnitude based pruning)

Figure 6: Signal propagation measurments (all signular value statistics) for the magnitude based pruning (Mag) on the 7-layer linear and tanh MLP networks. As described in the experiment settings in Appendix B, the magnitude based pruning is performed on a pretrained model. Notice that unlike other cases where pruning is done at initialization (*i.e.*, using either random or connection sensitivity based pruning methods), the singular value distribution changes abruptly when pruned (*i.e.*, note of the sharp change of singular values from 0 to 10% sparsity). Also, the singular values are not concentrated (note of high standard deviations), which explains rather inferior trainability compared to other methods. We conjecture that naively pruning based on the magnitude of parameters in a single-shot, without pruning gradually or employing some sophisticated tricks such as layerwise thresholding, can lead to a failure of training compressed networks.

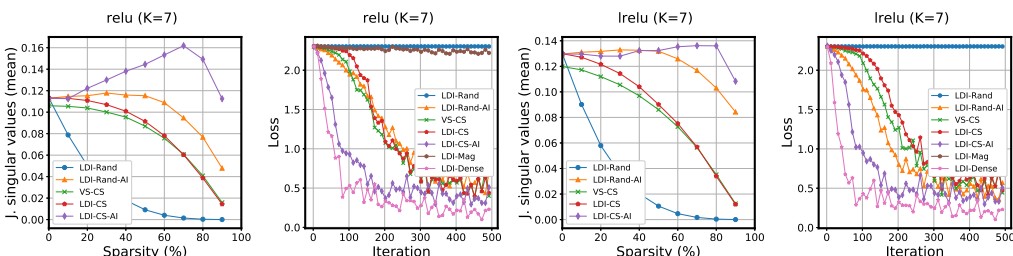

Figure 7: Signal propagation and training behavior for ReLU and Leaky-ReLU activation functions. They resemble those of the tanh case as in Figure 2, and hence the conclusion holds about the same.

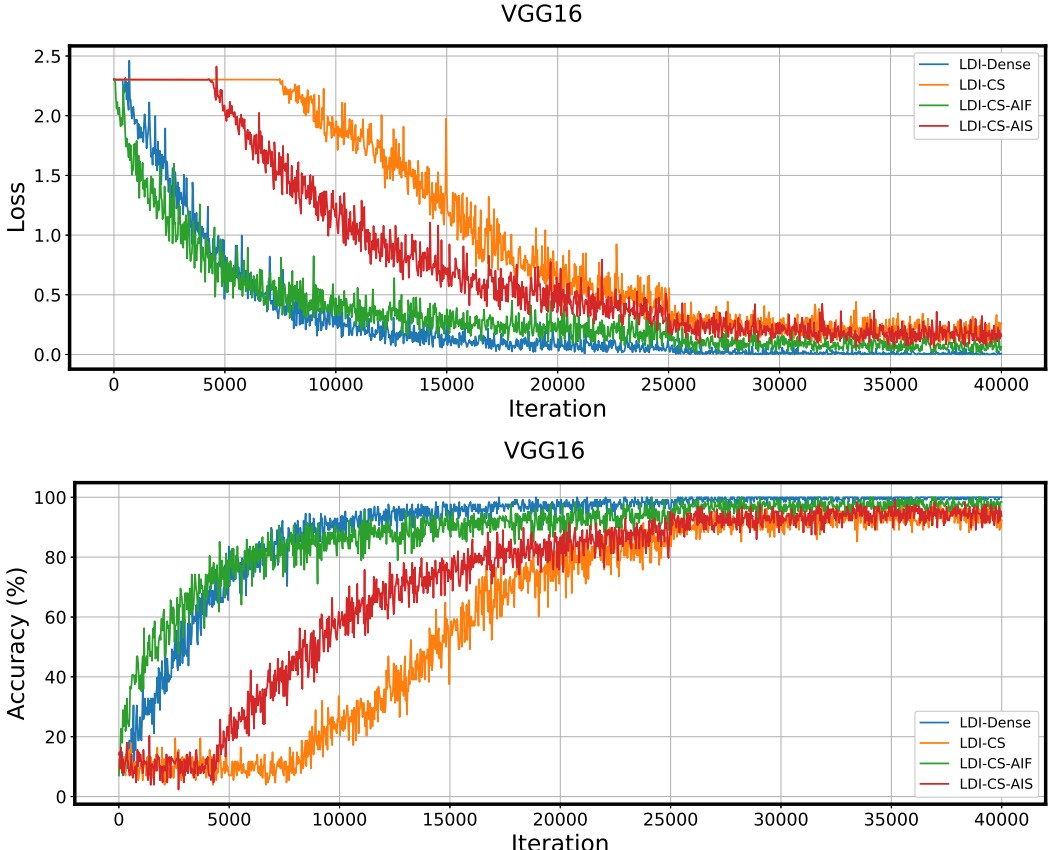

Figure 8: Training performance (loss and accuracy) by different methods for VGG16 on CIFAR-10. To examine the effect of initialization in isolation on the trainability of sparse neural networks, we remove batch normalization (BN) layers for this experiment, as BN tends to improve training speed as well as generalization performance. As a result, enforcing approximate isometry (LDI-CS-AIF) improves the training speed quite dramatically compared to the pruned network without isometry (LDI-CS). We also find that even compared to the non-pruned dense network (LDI-Dense) which is ensured layerwise dynamical isometry, LDI-CS-AIF trains faster in the early training phase. This result is quite promising and more encouraging than the previous case of MLP (see Figures 2 and 7), as it potentially indicates that an underparameterized network (by connection sensitivity pruning) can even outperform an overparameterized network, at least in the early phase of neural network training. Furthermore, we add results of using the spectral norm in enforcing approximate isometry in Equation 8 (LDI-CS-AIS), and find that it also trains faster than the case of broken isometry (LDI-CS), yet not as much as the case of using the Frobenius norm (LDI-CS-AIF).

