# OpenReview forum: "A Signal Propagation Perspective for Pruning Neural Networks at Initialization"
_ICLR.cc/2020/Conference — Accept (Spotlight)_

### Official Review · AnonReviewer2 · 2019-10-18
**Official Blind Review #2**

**Rating:** 3

**Review:**

The paper introduces a signal propagation perspective for single-shot network pruning. Particularly, by achieving layerwise dynamical isometry with proper initialization scheme, the paper ensure the reliability of connection sensitivity pruning criteria, thereby improving the pruning results.

I’m inclined to reject this paper because (1) methods of dynamical isometry typically care about trainability of very deep network (e.g., 10,000 layers) and it has little to do with generalization performance which is what we really care about in network pruning; (2) experiments on signal propagation were done in small datasets and fully-connected layers without batch norm, which is not the standard setting we conduct network pruning. It’s unclear whether the analysis would carry over to CNNs with batch norm.

Main argument:
This paper attacks single-shot pruning problem from the perspective of signal propagation. By using special initialization scheme, the authors attempt to calibrate the connection sensitivity measurements across layers. The main argument in the paper of dynamical isometry doesn’t has a direct connection with performance of a pruned network since it basically concerns the trainability of very deep neural networks.

Moreover, adopting orthogonal initialization satisfying layerwise dynamical isometry only leads to marginal improvements over standard variance scaling initialization, as shown in Table 2 and 3.

In terms of potential improvements, I would suggest the authors to first carefully study the relationship between trainability and generalization and also analyze signal propagation of networks with batch norm. Finally, for such an empirical paper, I expect more large-scale experiments conducted.


**Experience Assessment:**

I have published one or two papers in this area.

**Review Assessment: Checking Correctness Of Derivations And Theory:**

N/A

**Review Assessment: Checking Correctness Of Experiments:**

I assessed the sensibility of the experiments.

**Review Assessment: Thoroughness In Paper Reading:**

I read the paper at least twice and used my best judgement in assessing the paper.

---

> ### Author Response · Authors · 2019-11-12
> **Response to R2**
>
>
> We thank the reviewer (R2) for providing the feedback. We address the comments below.
>
>
> # Generalization performance
>
> As researchers we study many aspects (besides generalization) that help us better understand deep learning models. Therefore, we disagree with R2's assessment that we (in studying network pruning) care only about generalization. We elaborate below.
>
> > Trainability is important.
> Pruning neural networks at initialization, followed by training the sparse networks from scratch, is important and has great potential [B1]. However, finding a generalizable sparse network or training a sparse network from scratch is difficult as of yet [B2, B3]. In this work, we made a reasonable attempt to complement on-going investigations on the optimization difficulties of sparse networks by approaching from the perspective of signal propagation. We identified that a difficulty in pruning at initialization comes from degradation of signal propagation, and provided compelling evidence to support that the trainability of pruned networks is highly correlated with spectral measures of the networks' Jacobians, which is considered as an important, novel contribution by R3.
>
> > Network pruning is more than about generalization.
> Acceleration of training by improving signal propagation via enforcing approximate isometry reduces the total number of floating point operations for training a model (i.e., {# parameters to update} x {# training iterations}) quite significantly, compared to typical pruning-{after/while}-training approaches. This is useful in scenarios where high computational efficiency is required (e.g., distributed/decentralized optimization).
>
> > Generalization is not the main contribution or focus of this work.
> In this work, we show that the signal propagation perspective is useful for theoretically understanding and empirically improving the strategy of pruning neural networks at initialization. Specifically, we derived a theoretical characterization of the initialization condition to ensure faithful signal propagation before pruning (i.e., layer-wise dynamical isometry), demonstrated that pruning breaks dynamical isometry and degrades signal propagation on the pruned sparse networks (Fig. 2), proposed a data-free spectral method (Eq. (8)) to recover signal propagation (i.e., approximate dynamical isometry) improving training performance (Fig. 2), and further extended the strategy to other scenarios in practice (Section 5). While generalization is definitely an important concept in network pruning, a comprehensive study of generalization of sparse networks is beyond the scope of our work.
>
> > We nevertheless find a strong correlation between signal propagation and generalization.
> Our approach (i.e., LDI-AI) achieves the best generalization performance (e.g., compared to the average of VS methods, LDI-AI improves 0.24/0.55/0.70/0.35/1.30% for vgg16/resnet32/resnet56/resnet110/wrn16; see Table 2). While we do not claim any causality, we nevertheless find a strong correlation between signal propagation and generalization of sparse networks, which is robust across different architectures, datasets, and activation functions tested in this work, as observed by R3.
>
>
> # CNNs with batch norm
>
> Batch normalization (BN) tends to improve accuracy and speed up training [B4], and yet, there remains little consensus on the exact reason and mechanism behind these improvements [B5]. While understanding BN is an active research topic, as such, we remove BN and control the experiments in this work, such that we can focus on examining the effect of initialization for signal propagation and trainability of sparse networks. We note that this is the standard setup used in previous works on dynamical isometry and mean field theory [B6, B7], and further clarify that BN itself can cause gradient explosion [B8] which is critical to theoretical understanding of signal propagation derived in this work. While BN is pivotal in modern neural networks and has a practical value, understanding sparse networks under the influence of BN is beyond the scope of this work.
>
>
> References
> [B1] SNIP: Single-shot network pruning based on connection sensitivity, Lee et al. (ICLR 2019)
> [B2] The lottery ticket hypothesis: Finding sparse, trainable neural networks, Frankle and Carbin (ICLR 2019)
> [B3] The difficulty of training sparse neural networks, Evci et al. (ICML 2019 workshop)
> [B4] Batch normalization: Accelerating deep network training by reducing internal covariate shift, Ioffe and Szegedy (ICML 2015)
> [B5] Understanding batch normalization, Bjorck et al. (NeurIPS 2018)
> [B6] Exact solutions to the nonlinear dynamics of learning in deep linear neural networks, Saxe et al. (ICLR 2014)
> [B7] Exponential expressivity in deep neural networks through transient chaos, Poole et al. (NIPS 2016)
> [B8] A mean field theory of batch normalization, Yang et al. (ICLR 2019)

---

### Official Review · AnonReviewer3 · 2019-10-22
**Official Blind Review #3**

**Rating:** 8

**Review:**

This paper analyzes how signals propagate through randomly initialized neural networks that have undergone a kind of pruning/sparsification. The pruning method utilizes a metric called 'connection sensitivity', which has been used in prior work and which measures the infinitessimal impact of turning off specific parameters. The distribution of singular values in the layer-to-layer Jacobian matrices for pruned networks becomes increasingly pathological as the depth increases. This observation motivates the concept of 'layerwise dynamical isometry' (LDI), a slight generalization of the concept of 'dynamical isometry' that has been studied in prior work. Several methods for approximately obtaining differing amounts of LDI are investigated in a series of in-depth experiments that show a strong correlation between increased signal propagation and improved trainability of sparse networks.

Although there have been numerous works studying dynamical isometry as a principle for initializing very deep networks, and many other works studying pruning methods after (and recently before) training, as far as I'm aware there has been no prior work that examines the intersection of these two directions. As such, I found the contributions of this paper to be novel and believe the results will be of interest to practitioners and theorists alike.

An important contribution of this paper is in identifying that a main difficulty in pruning networks at initialization comes from degradation of signal propagation, leading to poor or impossible training. The numerous well-thought-out experiments provide compelling evidence that the trainability of pruned networks is highly correlated with spectral measures of the networks' Jacobians.

The authors take this observation a step further by introducing a method for correcting the poor conditioning that can result from pruning. They show that enforcing Approximate Isometry on weights of the pruned connectivity pattern enables the pruned models to train much faster and often achieve better performance.

Finally, the authors look at two natural extensions of their analysis  to designing new high-performing architectures to situations where labels are not present. Overall, I found this paper to have a detailed and thorough experimental analysis and to present nice new perspectives on pruning and signal propagation.

**Experience Assessment:**

I have published in this field for several years.

**Review Assessment: Checking Correctness Of Derivations And Theory:**

I assessed the sensibility of the derivations and theory.

**Review Assessment: Checking Correctness Of Experiments:**

I assessed the sensibility of the experiments.

**Review Assessment: Thoroughness In Paper Reading:**

I read the paper thoroughly.

---

> ### Author Response · Authors · 2019-11-12
> **Response to R3**
>
>
> We thank the reviewer (R3) for clearly summarizing the paper and appreciating our contributions.
>
> We are similarly excited by the comment "the trainability of pruned networks highly correlated with spectral measures of the networks' Jacobians, and hence, enforcing approximate isometry on weights of the pruned connectivity pattern enables the pruned models to train much faster and often achieve better performance" and the ongoing research in the community on this idea.
>
> We will attempt to reflect all reviewers’ comments on the final manuscript.

---

### Official Review · AnonReviewer1 · 2019-10-23
**Official Blind Review #1**

**Rating:** 6

**Review:**

In this paper, the authors studied and formalized the effect of initialization to connection-sensitivity-based pruning. The authors first pointed out that a previously studied pruning criterion -- connection sensitivity (CS) -- is a normalized magnitude of gradients. Based on signal propagation theory, to achieve a 'faithful' (with minimal amplification) CS, the gradients must be also faithful. Then by using relation of Jacobians and gradient, the authors proved that orthogonally initial weights guarantees faithful on linear networks and certain distribution property on nonlinear network can achieve layerwise dynamic isometry, which is to ensure faithful signal propagation. Based on these findings, the authors proposed an initialization setup for improving pruning performance, with the goal to ensure dynamic isometry by orthogonal initialization and approximation.

In summary, this paper used a typical pruning criterion (CS) and used signal propagation theory to studied how initialization effect CS. Although this linking is quite limited, this paper has marched a step in quantitative and theoretical study of initialization in pruning: how pruning affects network performance and why it is.

Here is a question:
In Fig.2, nonlinear activation is only shown with tanh. How is training performance when relu is used since relu is more commonly used in modern deep architectures.


**Experience Assessment:**

I have published one or two papers in this area.

**Review Assessment: Checking Correctness Of Derivations And Theory:**

N/A

**Review Assessment: Checking Correctness Of Experiments:**

I assessed the sensibility of the experiments.

**Review Assessment: Thoroughness In Paper Reading:**

I read the paper at least twice and used my best judgement in assessing the paper.

---

> ### Author Response · Authors · 2019-11-12
> **Response to R1**
>
>
> We thank the reviewer (R1) for the positive and constructive feedback.
>
> # ReLU
> We ran the same experiments for ReLU and Leaky-ReLU networks, and found that the same conclusion holds. The results are added to Figure 7 in Appendix C, and the key points are summarized as follows:
> - Pruning breaks dynamical isometry, and the more a network is pruned, the weaker signal propagation becomes on the pruned sparse network.
> - The better a network propagates signals, the faster it converges during training.
> - Enforcing approximate dynamical isometry recovers signal propagation on sparse networks, which in turn improves the training performance (rate of convergence) of sparse networks quite significantly.
> - In addition to signal propagation, the structure (the choice of pruning method) and the number of parameters (sparsity level) also affect trainability of sparse neural networks.
>
> In short, we find that signal propagation is an important property for understanding and improving trainability of sparse neural networks, and this applies to non-linear activation functions such as (Leaky-)ReLU.

---

### Decision · Program_Chairs · 2019-12-19

**Decision:**

Accept (Spotlight)

**Comment:**

This is a strong submission, and I recommend acceptance. The idea is an elegant one: sparsify a network at initialization using a distribution that achieves approximate orthogonality of the Jacobian for each layer. This is well motivated by dynamical isometry theory, and should imply good performance of the pruned network to the extent that the training dynamics are explainable in terms of a linearization around the initial weights. The paper is very well written, and all design decisions are clearly motivated. The experiments are careful, and cleanly demonstrate the effectiveness of the technique. The one shortcoming is that the experiments don't use state-of-the-art modern architectures, even thought that ought to have been easy to try. The architectures differ in ways that could impact the results, so it's not clear to what extent the same principles describe SOTA neural nets. Still, this is overall a very strong submission, and will be of interest to a lot of researchers at the conference.